Comparative mitogenomics of Cheiracanthium species (Araneae: Cheiracanthiidae) with phylogenetic implication and evolutionary insights

Li Zhaoyi 1
Zhang Feng dudu06042001@163.com 1 2
1 Key Laboratory of Zoological Systematics and Application of Hebei Province, College of Life Sciences, Hebei University , Baoding , Hebei , China
2 Hebei Basic Science Center for Biotic Interaction, Hebei University , Baoding , Hebei , China
Zhang Lin
Electronic publication date: 2025 Feb 14
Publication date: 2025
Volume: 13
Electronic Location ID: e18314
Received 2024 May 10; Accepted 2024 Sep 24
Copyright: ©2025 Li and Zhang
Copyright year: 2025
Copyright holder: Li and Zhang
License: This is an open access article distributed under the terms of the Creative Commons Attribution License, which permits unrestricted use, distribution, reproduction and adaptation in any medium and for any purpose provided that it is properly attributed. For attribution, the original author(s), title, publication source (PeerJ) and either DOI or URL of the article must be cited.
License URL: https://creativecommons.org/licenses/by/4.0/

Keywords: Cheiracanthiids, Divergence time, Mitochondrial genome, Molecular phylogenetics, Paraphyly

Funding: National Natural Science Foundation of China No. 32170468 Science and Technology Fundamental Resources Investigation Program No. 2022FY202100 This work was supported by the National Natural Science Foundation of China (No. 32170468), and by the Science and Technology Fundamental Resources Investigation Program (Grant No. 2022FY202100). The funders had no role in study design, data collection and analysis, decision to publish, or preparation of the manuscript.

==============================
The genus Cheiracanthium C. L. Koch, 1839 is the most species-rich genus of the family Cheiracanthiidae. Given the unavailability of information on the evolutionary biology and molecular taxonomy of this genus, here we sequenced nine mitochondrial genomes (mitogenomes) of Cheiracanthium species, four of which were fully annotated, and conducted comparative analyses with other well-characterized Araneae mitogenomes. We also provide phylogenetic insights on the genus Cheiracanthium. The circular mitogenomes of the Cheiracanthium contain 37 genes, including 13 protein-coding genes (PCGs), 22 transfer RNA genes (tRNAs), two ribosomal RNA genes (rRNAs) and one putative control region (CR). All genes show a high A+T bias, characterized by a negative AT skew and positive GC skew, along with numerous overlapped regions and intergenic spacers. Approximately half of the tRNAs lack TΨC and/or dihydrouracil (DHU) arm and are characterized with unpaired amino acid acceptor arms. Most PCGs used the standard ATN start codons and TAR termination codons. The mitochondrial gene order of Cheiracanthium differs significantly from the putative ancestral gene order (Limulus polyphemus). Our novel phylogenetic analyses infer Cheiracanthiidae to be the sister group of Salticidae in BI analysis, but as sister to the node with Miturgidae, Viridasiidae, Corinnidae, Selenopidae, Salticidae, and Philodromidae in ML analysis. We confirm that Cheiracanthium is paraphyletic, for the first time using molecular phylogenetic approaches, with the earliest divergence estimated at 67 Ma. Our findings enhance our understanding of Cheiracanthium taxonomy and evolution.

Introduction

Spiders (Araneae) are the second largest group within Arachnida, with over 50,000 species of 134 described families, occupying almost all terrestrial and some aquatic habitats (World Spider Catalog, 2024). As dominant arthropod predators in terrestrial ecosystems, spiders have adapted to various environments and developed a wide array of remarkable characteristics throughout their evolutionary history, including body shapes, behaviors, web architectures, respiratory systems, and venom compounds (Platnick, 2020). Rapid development of phylogenomic approaches using genomic-scale data have been increasingly applied to the systematics and evolutionary studies of spiders, including transcriptome data (Fernández et al., 2018), mitogenomic (Tyagi et al., 2020; Li et al., 2022), and ultra-conserved elements (Kulkarni et al., 2021; Kulkarni, Wood & Hormiga, 2023). The phylogenetic relationships of the major clades in the spider tree of life have been largely resolved and are well-supported across different data types.

Mitochondria are crucial cytoplasmic organelles in eukaryotic cells, playing pivotal roles in cell metabolism, disease, apoptosis, and senescence. The typical mitochondrial genome (mitogenome) of metazoa is a circular double-stranded DNA molecule ranging from 14 to 19 kb in size. It encodes 37 genes, including 13 protein-coding genes (PCGs), two ribosomal RNA (rRNA) genes, 22 transfer RNA (tRNA) genes, and a large non-coding region known as the control region (Boore, 1999; Cameron, 2014). Mitogenomes have provided valuable insights into the phylogenetic relationships, population genetics and evolution of various organisms (Wang et al., 2019; Ge et al., 2022; Zhang et al., 2022; Ye et al., 2024; Chen et al., 2024) including the spiders (Tyagi et al., 2020; Li et al., 2021; Li et al., 2022; Prada et al., 2023), and are characterized by a high nucleotide substitution rate, variable gene order, conservative gene content, maternal inheritance, and absence of introns (Castellana, Vicario & Saccone, 2011). Currently, more than 360 mitogenomes of spiders have been deposited in the GenBank database (https://www.ncbi.nlm.nih.gov/). Spider mitogenomes range in size from 13,211 bp in Atypus largosaccat to 16,000 bp in Argyroneta aquatic (Li et al., 2022), exhibiting variations in mitogenomic characteristics such as gene content, gene arrangement, and tRNA structure (Lavrov, Boore & Brown, 2000; Wang et al., 2016; Li et al., 2022). However, approximately one-sixth of the known spider mitogenomes belong to the family Araneidae, so the data on mitogenomes for other spider families remains relatively scarce, including the family Cheiracanthiidae.

Cheiracanthiids, including 14 genera and 372 valid species, are essentially wandering, nocturnal spiders serving as significant generalist predators of small invertebrates (World Spider Catalog, 2024). Currently, only two mitogenomes (Cheiracanthium triviale and C. erraticum) of the Cheiracanthiidae have been published (Tyagi et al., 2020; Li et al., 2022), and the limited available data have hindered comprehensive studies on this lineage. The genus Cheiracanthium represents the most species-rich lineage within Cheiracanthiidae, comprising 222 species (World Spider Catalog, 2024). This genus has often been considered paraphyletic based on morphological characteristics of the palps and epigyne (Wunderlich, 2012; Marusik & Fomichev, 2016; Esyunin & Zamani, 2020; Li & Zhang, 2023; Li & Zhang, 2024), but this proposed paraphyletic status has yet to be tested by molecular data.

In this study, we sequenced nine Cheiracanthium mitogenomes and successfully annotated the four complete mitogenomes: Cheiracanthium brevispinum Song, Feng & Shang, 1982; C. insigne O. Pickard-Cambridge, 1874; C. pichoni Schenkel, 1963; and C. solidum Zhang, Zhu & Hu, 1993. These complete mitogenomes were analyzed for gene content and composition, codon usage, evolutionary patterns, RNA secondary structure, and gene arrangement. Furthermore, to comprehensively explore the phylogeny and evolution of cheiracanthiid spiders, we assembled 30 spider mitogenomes from raw reads of ultra-conserved elements (UCEs) and transcriptome data available in the NCBI public database, including 12 cheiracanthiids (10 of Cheiracanthium; one of Macerio Simon, 1897; and one of Eutittha Thorell, 1878). Additionally, we obtained two Cheiracanthium mitogenomes and 26 other spider mitogenomes from the NCBI public database. Combining with our nine newly sequenced species, we reconstructed the phylogenetic tree using all available PCGs of 67 species to infer the evolutionary relationships and divergence time of chericanthiid spiders.

Materials and Methods

Sampling, DNA extraction, and sequencing

Specimens were collected in China (Table S1). All specimens were stored in 95% ethanol and deposited in the Museum of Hebei University, Baoding, China (MHBU). The genomic DNA was extracted with the DNeasy Blood and Tissue Kit (Qiagen, Hilden, Germany) following the manufacturer’s instructions, and 2 µL of RNase A (Solarbio, Beijing, China) was added to the DNA extraction and then left at room temperature for 2 min to remove RNA. The quantity of DNA was checked using a Qubit™ fluorometer. The library preparation was conducted using the NEXTFLEX Rapid DNA-Seq Kit 2.0 and the NEXTFLEX Unique Dual Index Barcodes (Set C) (Bioo Scientific, Austin, TX, USA) following the protocols by Zhang et al. (2023a). The libraries were then sent to Novogene Co. Ltd for sequencing using the Illumina NovaSeq platform with 150-bp paired-end reads.

Mitochondrial genome assembly and annotation

Thirty Araneae UCEs and transcriptomes, were downloaded from NCBI (https://www.ncbi.nlm.nih.gov/sra/). Quality trimming of raw reads of UCEs, transcriptomes, and newly sequenced species was completed using bbduk.sh (BBTools) (Bushnell, 2014) to filter the reads shorter than 15 bp or with more than five Ns, as well as trim the poly-A or poly-T tails of at least 10 bp. The remaining cleaned data were used to assemble the mitogenome using MitoZ 1.04 (Meng et al., 2019). Genome annotation was first performed in MitoZ and then further polished in the MITOS web-server (http://mitos2.bioinf.uni-leipzig.de) (Bernt et al., 2013), followed by manual check using Geneious 6.1.7 (Kearse et al., 2012).

Mitochondrial genome sequence analysis

The analyses of mitogenome characteristics were conducted on the four complete annotated Cheiracanthium mitogenomes.

The mitogenome map was drawn using Proksee web-server (https://proksee.ca). The secondary structures of the tRNA genes were predicted with MITOS web-server. The two leucine and two serine tRNA genes were further differentiated by numerals, where Leu1 = tRNALeu(CUN), Leu2 = tRNALeu(UUR), Ser1 = tRNASer(AGN), Ser2 = tRNASer(UCN).

The nucleotide composition, codon usage of PCGs, and relative synonymous codon usage (RSCU) were analyzed using PhyloSuite 1.2.3 (Zhang et al., 2020). AT and GC skew were calculated by the following formulae: AT skew = (A–T)/(A+T) and GC skew = (G–C)/(G+C) (Perna & Koeher, 1995). The nucleotide diversity (Pi) was estimated in DnaSP (Librado & Rozas, 2009). KaKs_Calculator v3.0 (Zhang, 2022) was used to calculate the nonsynonymous substitution rates (Ka), synonymous substitution rates (Ks) and Ka/Ks of each PCG with the with Model Averaging (MA) method. A Ka/Ks ratio of = 1, <1, or >1 in protein-coding sequences were interpreted as evolving under a neutral selection, a negative (purifying) selection, or a positive (diversifying) selection, respectively (Hurst, 2002; Zhang et al., 2006). The genetic distances were computed using MEGA 7.0 (Kumar, Stecher & Tamura, 2016) and applying the Kimura-2-parameter model.

Gene arrangement analysis

After fully annotating the mitogenomes of four Cheiracanthium species, we compared their gene order with those of available complete spider mitogenomes (Table S2) and the putative ancestral arthropod mitogenome (Limulus polyphemus; Lavrov, Boore & Brown, 2000). Gene order was determined using PhyloSuite v1.2.3 and visualized with iTOL (http://itol.embl.de/). For visualization purposes, we arbitrarily designated the start of the cox1 gene as position 1 in each genome, with the direction pointing towards cox2.

Phylogenetic analyses

A total of 67 Araneae mitogenomes (nine obtained in this work) were included in our phylogenetic analyses (Table S2). The two Liphistiid species (Liphistius erawan and Songthela hangzhouensis) were used as outgroups. PCGs were extracted using PhyloSuite, and aligned using Mafft 7.313 (Katoh & Standley, 2013) with the L-INS-I strategy. Ambiguously aligned areas were removed using trimAl 1.3 (Capella-Gutierrez, Silla-Martinez & Gabaldon, 2009). The individual gene alignments were concatenated using PhyloSuite. The best partitioning scheme and nucleotide substitution models were estimated with PartitionFinder2 in PhyloSuite using the Akaike information criterion (AICc) for maximum likelihood (ML) and Bayesian inference (BI) analyses. ML analysis was performed in IQ-TREE 2.2.0 (Nguyen et al., 2015) using the optimized model and partition scheme, and an ultrafast bootstrap analysis with 5,000 replicates was conducted to assess the node support. BI analysis was performed using MrBayes 3.2 (Ronquist et al., 2012) with four chains (one cold chain and three heated chains). Two independent runs of 2 million generations were carried out sampling every 1,000 generations with the first 25% of trees discarded as burn-in. The resulting trees were visualized via Figtree 1.4.4 (http://tree.bio.ed.ac.uk/software/figtree/).

Divergence time estimation

The divergence time was estimated using BEAST 2.6.3 (Bouckaert et al., 2014) based on 13 concatenated PCGs of 67 spiders. A BEAST XML file was generated in BEAUTi 2, using the GTR model, and the gamma parameter set to 4. We assumed a Yule speciation process for the tree prior and a relaxed log-normal distribution for the molecular clock model. Fossil calibration points were taken from a recent spider fossil review (Magalhaes et al., 2020); five fossils were employed to calibrate the phylogenetic tree: Palaeothele montceauensis (299–304 Ma) for the Mesothelae stem, Eoplectreurys gertschi (164–175.1 Ma) for the Synspermiata stem, Montsecarachne amicorum (125–129.4 Ma) for the Synspermiata crown, Oxyopes succini (43–47.8 Ma) for the Oxyopidae stem, and Almolinus ligula (43–47.8 Ma) for the Salticidae crown. However, no fossils were available for calibrating the cheiracanthiids clade in the present study. The branch lengths were transformed to ultrametric in Mesquite 3.04 (Maddison & Maddison, 2015) before conducting divergence dating analyses. Two independent Markov Chain Monte Carlo (MCMC) runs were performed for 500 million generations, sampling every 1,000 generations. Tracer 1.7.1 (Rambaut et al., 2018) was used to assess convergence and ensure that the effective sample size (ESS) was greater than 200 for all parameters. Post-burn-in trees (with the first 25% of samples discarded as burn-in) and their parameters were summarized using TreeAnnotator 1.4.7 (Bouckaert et al., 2014) to generate a maximum clade credibility (MCC) chronogram. The MCC chronogram showed the mean divergence time estimates with 95% highest posterior density (HPD) intervals. Finally, the chronogram was visualized and edited using Figtree.

Results

Mitogenome organization and nucleotide composition

We completely annotated the mitogenomes of four species of the genus Cheiracanthium: C. brevispinum (OQ559338, length: 14,703 bp), C. insigne (OP785748, length: 14,598 bp), C. pichoni (OR726571, length: 14,666 bp), and C. solidum (OR726570, length: 15,185 bp) (Fig. 1). Each mitogenome was a double-stranded circular DNA molecule, containing the typical metazoan mitochondrial gene set of 37 genes (13 PCGs, 22 tRNAs, and two rRNAs), along with a large non-coding control region. Among these genes, 22 are encoded on the major strand (J-strand), while the remaining 15 are on the minor strand (N-strand) (Table S3).

Figure 1 Mitochondrial maps of (A) C. brevispinum, (B) C. insigne, (C) C. pichoni, and (D) C. solidum.

Genes are represented by different colour blocks. The GC content is plotted using a black sliding window, as the deviation from the average GC content of the entire sequence. GC skew is plotted as the deviation from the average GC skew of the entire sequence.

The nucleotide composition of whole mitogenomes of each species was highly A+T biased (77.8% in C. brevispinum, 77.9% in C. insigne, 79.2% in C. pichoni, and 78.7% in C. solidum). Similarly, the A+T content of PCGs, tRNAs, and rRNAs is high, up to 78%, 77.8%, and 80% in C. brevispinum, 77.6%, 77.5%, and 81.4% in C. insigne, 78.7%, 80.8%, and 81.0% in C. pichoni, and 78.4%, 78.5%, and 81.8% in C. solidum, respectively. The whole mitogenomes of the four species exhibit a preference for T over A and G over C, with a negative AT skew but positive GC skew (−0.107 and 0.223 in C. brevispinum, −0.109 and 0.192 in C. insigne, −0.091 and 0.189 in C. pichoni, −0.085 and 0.199 in C. solidum, respectively) (Table S4).

In the mitogenomes of C. brevispinum, C. insigne, C. pichoni, and C. solidum (Table S3), 16, 15, 20, and 17 overlapping nucleotides were detected, respectively. These overlaps totaled 375 bp, 302 bp, 244 bp, and 228 bp, respectively. The longest overlap occurs between nad4 and nad4L in the mitogenomes of four species, length of 113 bp, 104 bp, 44 bp, and 44 bp, respectively. Intergenic spacers in the mitogenome were found in 12 locations in C. brevispinum (total size of 195 bp), 10 in C. insigne (105 bp), 11 in C. pichoni (140 bp), and 12 in C. solidum (128 bp), and the longest spacer was found between rrnL and trnV in C. brevispinum (74 bp), and between trnA and trnN in C. insigne (40 bp), C. pichoni (87 bp), and C. solidum (45 bp).

Protein coding genes (PCGs)

The total length of the 13 mitochondrial protein coding genes (PCGs) is 10,848 bp in C. brevispinum, 10,806 bp in C. insigne, 10,713 bp in C. pichoni, and 10,689 bp in C. solidum. Similar to the full mitogenome, all PCGs show a high A+T bias (78% in C. brevispinum, 77.6% in C. insigne, 78.7% in C. pichoni, and 78.4% in C. solidum). Furthermore, the A+T content at the third codon positions (91.7% in C. brevispinum, 91.7% in C. insigne, 93.5% in C. pichoni, and 92.6% in C. solidum) was significantly higher than the first (72%, 71.3%, 72.5%, and 72.0% in four species, respectively) and second (70.3%, 69.9%, 70.3%, and 70.4%) codon positions. Additionally, all PCGs display negative AT skewness and positive GC skewness (Table S4). Among the 13 mitochondrial PCGs of the four species, only four genes (nad5, nad4, nad4L, and nad1) were encoded on the N-strand, while the remaining genes were all encoded on the J-strand (Table S4). All 13 PCGs started with a typical ATN start codon, except for cox2 and cox3, which started with TTG. Notably, in C. brevispinum, nad4 was initiated by TTG. For stop codons, most PCGs terminated with TAR (TAA/TAG), while a few had an incomplete stop codon T-/TA- (Table S3).

The relative synonymous codon usage (RSCU) in 13 PCGs of the four Cheiracanthium species is presented in Fig. 2. Among these, the UUU (Phe), AUU (Ile), AUA Met), and UUA (Leu2) were the most frequently utilized codons. All synonymous codons ending with A or U are more frequent than those ending with C or G, reflecting a higher frequency of AT usage over GC usage in the third codon positions.

Figure 2 The relative synonymous codon usage (RSCU) of (A) C. brevispinum, (B) C. insigne, (C) C. pichoni, and (D) C. solidum.

Codon families are provided on the x-axis along with the different combinations of synonymous codons that code for that amino acid. RSCU (relative synonymous codon usage) is defined on the Y axis. The top number of columns indicates the occupancy (%) of each individual amino acid used. Absent codon is provided at the bottom of columns.

Nucleotide diversity of 13 PCGs among four Cheiracanthium species was assessed using sliding window analysis (Fig. 3). The average diversity ranged from 0.106 (cox1) to 0.330 (atp8). Notably, atp8 (Pi = 0.330) exhibited the highest variability, followed by nad6 (Pi = 0.239), nad 5 (Pi = 0.217), and nad2 (Pi = 0.207). In contrast, cox1 (Pi = 0.106), cox2 (Pi = 0.130), atp6 (Pi = 0.135), and cytb (Pi = 0.139) had relatively low values, indicating greater gene conservation. This result indicates that nucleotide diversity was highly variable among the 13 PCGs.

Figure 3 The nucleotide diversity (Pi) of 13 PCGs in four Cheiracanthium mitogenomes determined via sliding window analysis (sliding window: 100 bp; step size: 25 bp), the average Pi value of each gene is shown under the gene name.

The average pairwise genetic distances among the mitogenomes of four Cheiracanthium species is summarized in Fig. 4A. Genes that evolved relatively fast are atp8 (0.448), nad6 (0.291), nad5 (0.257), and nad2 (0.243), those that evolved more slowly are cox1 (0.114), cox2 (0.144), atp6 (0.149), and cytb (0.155). Furthermore, the evolutionary rate of 13 PCGs was estimated by analyzing Ka/Ks (Fig. 4B). The results show that atp8 has the highest mean Ka/Ks value, which indicates it may have evolved more rapidly than the other PCGs in Cheiracanthium, whereas cox2 has the lowest mean value of Ka/Ks, implying a slower rate of evolution. Comparing the mean Ka/Ks values of PCGs among Cheiracanthium species, the results show that the mean Ka/Ks values of all species are below 1.

Figure 4 Boxplots of (A) Ka/Ks and (B) genetic distances for the 13 PCGs of the four Cheiracanthium species.

Transfer and ribosomal RNA genes

The total lengths of tRNAs varied across the mitogenomes of C. brevispinum (1,288 bp), C. insigne (1,314 bp), C. pichoni (1,298 bp), and C. solidum (1,333 bp). Among the 22 tRNAs (Table S4), 13 were encoded on the J-strand, while nine were on the N-strand. The shortest tRNA was the trnG (40 bp) in C. insigne, while the longest was trnR (86 bp) in C. solidum. The concatenated sequence of all 22 tRNAs shows a high A+T bias and positive AT and GC skewness in all species, indicating a preference for A over T and G over C in the total tRNA pool.

The secondary structures of the 22 tRNAs of the four Cheiracanthium mitogenomes are illustrated in Figs. S1–S4, 1 for each amino acid, with an additional isoform for each of serine and leucine. Nearly half of the tRNAs in each species lack either the dihydrouracil (DHU) stem or the TψC arm, appearing as simplified loops rather than typical cloverleaf-shaped structures. These truncated tRNA genes may be products of overlap with adjacent genes. While Watson-Crick base pairings (A-T and G-C) are prevalent in most tRNAs, mismatched pairs (U-U, U-C, A-C, T-G, A-A, A-G, G-G, A-C, and C-C) are observed throughout the arms of almost all tRNAs in all species.

The two expected rRNAs (16S rRNA and 12S rRNA) of the four species are located on the N-strand, separated by a single trnV (Fig. 1, Table S3). The length of rrnL ranged from 917 bp (C. brevispinum) to 1,045 bp (C. solidum), while rrnS ranged from 683 bp (C. insigne) to 716 bp (C. brevispinum). Additionally, both rRNAs exhibit a positive AT skew and negative GC skew (Table S4).

Mitochondrial gene rearrangement

Our results (Fig. 5) showing that the gene orders of all PCGs were conserved within Araneae, with more variability observed in the position of tRNAs. Each lineage within Mesothelae, Mygalomorphae, Synspermiata, and Entelegynae exhibits a distinct major gene order. The gene order of the four Cheiracanthium species was consistent with that of most Entelegynae spiders, but distinctly different from those of Mesothelae, Mygalomorphae, and Synspermiata spiders. In comparison with the putative ancestor (Limulus polyphemus) and Mesothelae species, nine gene transpositions were identified in Cheiracanthium, including trnL2UUR, trnN, trnA, trnS1AGN, trnR, trnI, trnT, trnY, and trnC. The transposition of trnL2 shifted from downstream of nad1 to upstream, creating new gene boundaries at nad3-trnL2. The trnT transposed after trnS2 and trnI inserted after nad6, contrasting with their initial locations after nad4L and rrnS in the putative ancestor, respectively. The transposition of trnA, trnR, trnN, trnS1AGN occurred from the original gene block trnA-trnR. The trnN-trnS1AGN resulted in new gene boundaries at trnN-trnA-trnS1AGN-trnR, trnC and trnY exchanged their positions.

Figure 5 Linearized comparison of the gene arrangement in major spider mitogenomes.

The colored blocks represent different categories of genes. “-” refers to genes on the minor strand (N-strand).

Phylogenetic and divergence time analyses

We primarily focused on exploring the phylogeny and evolution of cheiracanthiids, including 11 family members of Dionycha and other major spider families in our data. Phylogenetic trees from BI and ML based on the 13 PCGs of 67 species are summarized in Fig. 6 and are also presented in Figs. S5–S6.

Figure 6 Phylogenetic tree summarizing results from Bayesian inference (BI) and maximum likelihood (ML) approaches of 67 spider mitogenomes, based on nucleotide sequences of 13 PCGs.

Topology obtained in the BI analyses. The numbers at the nodes separated by “/” indicate the posterior probability (BI) and bootstrap value (ML). Nodes with a “X” indicate topologies not recovered in the ML analysis.

Both the BI and ML trees strongly supported all major lineages within Araneae, recovering a deep split between the two suborders, Mesothelae and Opisthothelae (Mygalomorphae and Araneomorphae). The diverse infraorder Araneomorphae encompasses Synspermiata, Araneoidea, and the RTA clade. The family Cheiracanthiidae was clustered within the Dionycha clade, which is placed as the sister group to Salticidae in the BI analysis (posterior probability (pp) = 0.98), while the sister group to the node with Miturgidae, Viridasiidae, Corinnidae, Selenopidae, Salticidae, and Philodromidae in the ML analysis (bootstrap (bs) = 99%). Within Cheiracanthiidae, Macerio was identified as sister to the remaining cheiracanthiid species, receiving strong (pp = 1, bs = 100%). Cheiracanthium formed a paraphyletic group in BI and ML analyses due to Eutittha being nested within it, but with a relatively low support (pp = 0.86, bs = 84%).

The ML and BI analyses yielded slightly different relationships within Cheiracanthium. For example, in the BI analysis, the tree was resolved as ((C. mildei_ZY05, C. sp_ZY07), (C. sp_ZYL314, (C. exquestitum_ZY203, ((C. pichoni, C. sp_ZY09), (C. eutittha_ZY01, C. sp_ZY08))))) with a posterior probability (pp) of 0.9, whereas the ML analysis produced a tree with the structure ((C. mildei_ZY05, C. sp_ZY07), ((C. taegense_ZYL137, (C. solidum, C. digitatum_ZYL318)), (C. unicum_ZY10, (C. sp_ZY06, (C. insigne, C. triviale))))) but with lower support (bs = 48%).

Based on the fossil-calibrated phylogeny (Fig. 7), Cheiracanthiidae diverged from the sister-group (Salticidae) around 102 Ma (95% confidence interval: 87–117 Ma). The first divergence within Cheiracanthiidae occurred in 87 Ma (75–101 Ma). Subsequently, rapid divergence occurred within Cheiracanthium, approximately 67 Ma (57–78 Ma).

Figure 7 Divergence time estimation of the 67 spiders.

Node bars indicate 95% confidence intervals of the divergence time estimate. Chronostratigraphic scale data are derived from International Commission on Stratigraphy (https://stratigraphy.org/).

Discussion

Mitochondrial genome characterization

The sizes and content of the four completely annotated Cheiracanthium mitogenomes are highly conserved and fall well within the range observed in published spider mitogenomes (Li et al., 2022). There is no discernible expansion or contraction of the mitogenomes within Cheiracanthium species during the diversification process. The mitogenomes of these four species are biased in nucleotide composition ((A+T)% >(G+C)%) and often prefer T over A and C over G, resulting in a negative AT skew, positive GC skew (Table S4). This pattern of nucleotide skewness in mitogenomes is commonly observed in other chelicerates (Masta, Longhorn & Boore, 2009), and is consistent with other Opisthothele spiders, whereas Mesothele spiders show a different pattern with a positive or negative AT skew and negative GC skew (Li et al., 2022). The positive AT skew and negative GC skew pattern is also shown in the mitogenome of Limulus polyphemus, an ancient arthropod whose mitogenome is considered a putative ancestral pattern for chelicerates (Staton, Daehler & Brown, 1997; Lavrov, Boore & Brown, 2000). These results indicate that a reversal bias in nucleotide composition likely emerged after the divergence of Opisthothele spiders from their common ancestor with Mesothele spiders, and has since been retained throughout the evolutionary history of Opisthothele spiders.

Although mitogenomes are generally compact, overlapping genes and intergenic spaces are common in arthropod mitogenomes (Mans et al., 2012; Wang et al., 2016; Zhu et al., 2018; Tyagi et al., 2020; Ge et al., 2022). The longest overlapping gene, located between nad4 and nad4L, is observed in all four Cheiracanthium species. These overlaps between genes may be a result of selective pressure to reduce genome size, contributing to the exceptional compactness of mitogenome organization (Ojala, Montoya & Attardi, 1981; Rand, 1993; Lavrov & Brown, 2001). Additionally, the longest intergenic spaces are found between rrnL and trnV in C. brevispinum, and between trnA and trnN in C. insigne, C. pichoni, and C. solidum. These long intergenic spaces were considered as the results of gene rearrangements (Wu et al., 2014).

For the PCGs, four Cheiracanthium species exhibit the same AT and GC skewness pattern as the full mitogenome, which may influence codon usage (Wang et al., 2016). The majority of PCGs commenced with common start codons ATN, which is a frequently used start codon in Araneae mitogenomes (Tyagi et al., 2020). Most PCGs terminated with TAR, while a few ended with incomplete stop codons T-/TA-, which are presumed to be completed via post-transcriptional polyadenylation and are commonly observed in metazoan mitogenomes (Ojala, Montoya & Attardi, 1981). Synonymous codons ending with A or U are more frequent than those ending with G or C, consistent with other Araneae mitogenomes (Kumar et al., 2020).

The atp8 and nad6 genes show relatively high nucleotide diversity (Pi), pairwise genetic distances, and Ka/Ks values compared to other genes. This suggests that these two genes may have evolved under relatively relaxed purifying selection, potentially related to adaptation to new environments (Nielsen, 2005). For instance, positive selection in atp8 has been implicated in the evolution of flight in bats to meet increased energy demands (Shen et al., 2010), while atp6 has been associated with galliforms’ adaptation to high altitudes (Zhou et al., 2014).

tRNAs are key in translation, acting as adapter between mRNA codons and amino acids (Lorenz, Lünse & Mörl, 2017). The typical set of 22 tRNAs was identified in the four Cheiracanthium species, which is consistent with other published Araneae mitogenomes (Li et al., 2022; Prada et al., 2023), the DHU arm and/or TψC arm are commonly absent and simplified to a loop, a common characteristic observed in metazoan mitogenomes (Garey & Wolstenholme, 1989; Park, Lee & Hwang, 2007; Masta & Boore, 2008; Zhang et al., 2023b). The presence of truncated mitochondrial tRNAs may cause challenges for accurately annotating specific tRNA genes (Tyagi et al., 2020). However, the truncated secondary structure of certain tRNAs may not affect the translation process. For example, previous research has demonstrated that mitochondrial tRNAs with extremely short structures in nematodes can still be recognized by synthesizing enzymes and undergo aminoacylation (Giegé et al., 2012). Additionally, numerous unmatched base pairs were identified in tRNA stems in the four Cheiracanthium species, which may represent a typical feature of spider mitochondrial tRNAs, as observed in other species such as Habronattus oregonensis (Masta & Boore, 2004), Carrhotus xanthogramma (Fang et al., 2016), Ebrechtella tricuspidate (Zhu et al., 2018), and Cheiracanthium triviale (Tyagi et al., 2020).

Comparative analyses of mitochondrial gene orders are a powerful method of revealing ancient events in the process of species evolution (Boore et al., 1995), such as two species of Mesothelae (Fig. 5) exhibit the same gene order as Limulus polyphemus, which is considered a putative ancestral pattern for arthropod gene arrangement (Staton, Daehler & Brown, 1997; Lavrov, Boore & Brown, 2000). However, Opisthothelae spiders share similar gene arrangement pattern, and with a low rate of rearrangement, but it differs significantly from that of Mesothelae (Fig. 5). Therefore, the suborders of Araneae can be distinguished by the mitogenomic gene order synapomorphies, as Boore (1999) suggested that gene rearrangements serve as useful markers for resolving deep splits within phylogenies. Within the genus Cheiracanthium, the gene order remains consistent across all examined species, whereas nine tRNAs have been found to have translocated positions compared to the putative ancestral arthropod mitogenome (Limulus polyphemus) and Mesothelae species. As reported in previous studies on mitogenomic gene arrangement, mitochondrial tRNAs undergo a high density of post-transcriptional modifications, which may cause frequent gene rearrangements (Lorenz, Lünse & Mörl, 2017).

Phylogenetic and divergence time analyses

The phylogenetic trees from the ML and BI analyses recovered the major lineage (Fig. 6), including Mesothelae, Mygalomorphae, Synspermiata, Araneoidea, and RTA clade, the relationships among them are consistent with previous studies using multiple loci (Wheeler et al., 2017), transcriptomes (Fernández et al., 2018), mitochondrial genes (Li et al., 2022; Prada et al., 2023), and UCEs (Kulkarni, Wood & Hormiga, 2023). Within the Dionycha clade, the phylogenetic position of Cheiracanthiidae shows conflicting yet highly supported relationships in both BI and ML analyses (pp = 0.98, bs = 99%; Figs. S5 and S6). This conundrum has affected phylogenetic studies across various organisms, including birds (Walker, Brown & Smith, 2018; Cloutier et al., 2019), placental mammals (Romiguier et al., 2013), and arachnids (e.g., Sharma et al., 2014; Ballesteros & Sharma, 2019; Lozano-Fernández et al., 2019). However, the previous phylogeny revealed a close association of Cheiracanthiidae with Philodromidae (Wheeler et al., 2017) or Selenopidae + Viridasiisae (Azevedo et al., 2022; Kulkarni, Wood & Hormiga, 2023) or Philodromidae + Salticidae (Arakawa et al., 2022). This discrepancy is largely attributed to the limited sampling of Cheiracanthiidae and other families included in phylogenetic studies.

An interesting finding from the phylogenetic analyses is the placement of Eutittha mordax, which was nested within Cheiracanthium (Fig. 6). Eutittha mordax was originally classified in the genus Cheiracanthium but was later transferred into Eutittha by Esyunin & Zamani (2020) based on morphological characteristics. We also agree with Esyunin and Zamani’s view that Cheiracanthium mordax is closely related to the Eutittha species due to the absence of a median apophysis (vs. present in Cheiracanthium). Our analyses confirm the paraphyletic status of Cheiracanthium based on molecular phylogenetic data, as Eutittha is nested within it, supporting previous morphological views proposed by Wunderlich (2012), Marusik & Fomichev (2016), Li & Zhang (2023), and Li & Zhang (2024).

The most significant divergences within hunting spiders occurred during the Cretaceous Terrestrial Revolution (Fig. 7; Shao & Li, 2018). For instance, the most recent common ancestor of cheiracanthiids diversified about 87 Ma (Fig. 7). Subsequently, rapid diversification within the genus Cheiracanthium occurred between 78–57 Ma, coinciding with major global warming events during the Late Cretaceous. Temperature fluctuations likely promoted the reorganization of biotic communities and speciation. (Vajda & Bercovici, 2014; Cui et al., 2019; Bondarenko & Utescher, 2022). For example, salamanders underwent rapid diversification and dispersal episodes that coincided with major global warming events during the Late Cretaceous and again during the Paleocene–Eocene thermal optimum (Vieites, Min & Wake, 2007), the diversification of all major lineages in the modern genus Prunus may have been triggered by the early Eocene global warming period (Chin et al., 2014). Therefore, the warm climatic conditions of the Cretaceous were crucial in shaping the extreme diversity of Cheiracanthium spiders. In addition, angiosperms underwent extensive radiation during the mid to late Cretaceous (Friis, Pedersen & Crane, 2006; Rudall, 2012), as did various plant-dependent insect lineages also began to rapidly diversify, including beetles (Mckenna et al., 2015), lepidopterans (Wahlberg, Wheat & Peña, 2013), ants (Moreau et al., 2006), and holometabolous insects (Misof et al., 2014). As key insect predators, spiders may also have diversified rapidly along with their prey (e.g., Peñalver, 2006; Selden & Penney, 2010). The significant increase in these insect groups may have favored spiders that prey on them and helps explain the rapid diversification of the Cheiracanthium at the Cretaceous–Paleogene boundary.

Conclusion

The complete mitogenomes of four Cheiracanthium spiders exhibit a typical circular molecule structure. The size of these mitogenomes range from 14,598 bp to 15,185 bp. All genes show a high A+T bias, with a negative AT skew and positive GC skew, but notably differ in base composition, skew values, and codon usage. Almost half of the tRNAs cannot fold into typical cloverleaf-shaped secondary structures, lacking either the TΨC and/or DHU arm and displaying simplified loops instead. Comparative analyses reveal significantly rearrangements in the gene orders of Cheiracanthium mitogenomes compared to the ancestral mitogenome (Limulus polyphemus). The family Cheiracanthiidae is closely related to Salticidae in the BI analysis, but as the sister group to the node with Miturgidae, Viridasiidae, Corinnidae, Selenopidae, Salticidae, and Philodromidae in ML analysis. Phylogenetic trees based on mitochondrial genome sequences indicate that Cheiracanthium is not monophyletic and show rapid divergence within this genus around 67 Ma.

Supplemental Information

Supplemental Information 1 Detailed collection information of nine newly sequence Cheiracanthium species

Supplemental Information 2 Information of the representative taxa and accession numbers for the mitogenome sequences, with summary on the assembly and annotation results

Accession numbers with an asterisk (*) indicate newly obtained mitochondrial genome sequences in this study, nine newly sequenced species are in bold.

Supplemental Information 3 Mitochondrial genome organization of Cheiracanthium brevispinum, C. insigne, C. pichoni, and C. solidum.

The numbers at the nodes separated by “/” indicate the C. brevispinum, C. insigne, C. pichoni, and C. solidum. IGN represents (+) values as intergenic nucleotides and (-) values as overlapping regions. J refers to the major strand; N refers to the minor strand.

Supplemental Information 4 Base composition and skewness of mitogenomes of Cheiracanthium brevispinum, C. insigne, C. pichoni, and C. solidum.

The numbers at the nodes separated by “/” indicate the C. brevispinum, C. insigne, C. pichoni, and C. solidum. J refers to the major strand; N refers to the minor strand.

Supplemental Information 5 Secondary structures of the 22 tRNA genes of Cheiracanthium brevispinum mitochondrial genome

The tRNAs are labeled with the abbreviations of their corresponding amino acids. Names of structural components of a tRNA gene are indicated in the trnY structure.

Supplemental Information 6 Secondary structures of the 22 tRNA genes of Cheiracanthium insigne mitochondrial genome

The tRNAs are labeled with the abbreviations of their corresponding amino acids.

Supplemental Information 7 Secondary structures of the 22 tRNA genes of Cheiracanthium pichoni mitochondrial genome

The tRNAs are labeled with the abbreviations of their corresponding amino acids.

Supplemental Information 8 Secondary structures of the 22 tRNA genes of Cheiracanthium solidum mitochondrial genome

The tRNAs are labeled with the abbreviations of their corresponding amino acids.

Supplemental Information 9 Phylogenetic tree of 67 spider mitogenomes using maximum likelihood (ML)

The relationships were constructed based on nucleotide sequences of 13 protein-coding genes. The numbers at the nodes are bootstrap values.

Supplemental Information 10 Phylogenetic tree of 67 spider mitogenomes using Bayesian inference (BI)

The relationships were constructed based on nucleotide sequences of 13 protein-coding genes. The numbers at the nodes are Bayesian posterior probabilities.

Supplemental Information 11 The sequences: PP626053 to PP626087

Supplemental Information 12 The sequence: OR726570

Supplemental Information 13 The sequence: OR726571

We are grateful to Zegang Feng, Long Lin, Yuhui Ding and Wenqiang Zhang (Hebei University) for their assistance with the data analysis, to the two reviewers for their valuable comments and suggestions to improve the manuscript.

Additional Information and Declarations

Competing Interests

Author Contributions

DNA Deposition

Data Availability

The authors declare there are no competing interests.

Zhaoyi Li performed the experiments, analyzed the data, prepared figures and/or tables, and approved the final draft.

Feng Zhang conceived and designed the experiments, authored or reviewed drafts of the article, and approved the final draft.

The following information was supplied regarding the deposition of DNA sequences:

The mitogenomic sequences used in this article are available at GenBank: PP626053 to PP626087, and OQ559338, OP785748, OR726571, OR726570.

The following information was supplied regarding data availability:

The data is available at NCBI SRA: SRR28892194 to SRR28892202.

The PCGs are available at figshare: Li, Zhaoyi (2025). 67spp_PCG of MS: “Comparative mitogenomics of Cheiracanthium species (Araneae: Cheiracanthiidae) with phylogenetic implication and evolutionary insights”. figshare. Dataset. https://doi.org/10.6084/m9.figshare.26877529.v2.

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
