# Peer review of "Comparative mitogenomics of Cheiracanthium species (Araneae: Cheiracanthiidae) with phylogenetic implication and evolutionary insights"

_PeerJ, doi:10.7717/peerj.18314_

## Round 0.1 · original submission · Minor Revisions

Thank you for your submission to PeerJ.

Please change as reviewers' comments.

·

Basic reporting

The language of this manuscript is clear, but there are some minor issues worth noting:
1. Line 33-34, this sentence is incomplete.
2. Line 36, "systemic" should be "phylogenetic", keeping consistent in all parts.
3. Line 46, maintaining consistency in context and tense. "confirmed" should be "confirm".
4, Line 81, remains.
5, Line 87, "and" should be "but".
6, Line 90, considered to be.
7, Line 93, "confirmed" should be "tested".
The other basic parts are well done, I have no comment on them.

Experimental design

The manuscript presents original primary research within the aims and scope of the journal, with a well-defined, relevant, and meaningful research question. It explicitly outlines how the research fills an identified knowledge gap, e.g., providing the molecular phylogenetic and evolutionary insights into the genus Cheiracanthium. The investigation is conducted rigorously, adhering to high technical and ethical standards. Additionally, the methods are described with sufficient detail and information to facilitate replication. The result advances our understanding of cheiracanthiids evolution and will have downstream benefits for studies of taxonomy and phylogeny.

Validity of the findings

All underlying data have been provided, ensuring robustness, statistical soundness, and control. The conclusions are well articulated, directly linked to the original research question, and confined to supporting the obtained results.

Additional comments

1, Line 437-438 “This discrepancy is largely due to the limited taxon sampling in this study, as few
cheiracanthiid spiders have been sequenced and included in phylogenetic analyses”. I think it is also resonable due to limited samples of other families.
2, I think it will be better to discuss more for the dating analysis which is perfectly done. Such as, why the rapid divergence happened in 67 Ma?

Reviewer 2 ·

Basic reporting

Clear, unambiguous, professional English language used throughout.
Intro & background to show context.
 Literature well referenced & relevant.
 Structure conforms to PeerJ standards, discipline norm, or improved for clarity.
 Figures are relevant, high quality, well labelled & described.
 Raw data: The authors should upload the PCGs that used in this article to the public platform, such as FigShare (https://figshare.com/) and then give us the DOI number in the Data Availability section.

Experimental design

Original primary research within Scope of the journal.
 Research question well defined, relevant & meaningful. It is stated how the research fills an identified knowledge gap.
 Rigorous investigation performed to a high technical & ethical standard.
 Methods: There some detailed information should be added, which was shown in the context.

Validity of the findings

All underlying data have been provided; they are robust, statistically sound, & controlled.
 Conclusions are well stated, linked to original research question & limited to supporting results.

Additional comments

(1) In the INTRODUCTION section, although it covers the basic researches related to the genus Cheiracanthium, yet there is a lack of citation of the latest studies about spider’s phylogeny.
(2) The Latin names should be italic in the main context, Figures, Tables, and all Additional files.
(3) The ethics approval should be provided.
(4) [IMPORTANT] The detailed revised information has been marked in the original text. Please read it carefully and make the necessary revisions.

Annotated reviews are not available for download in order to protect the identity of reviewers who chose to remain anonymous.

---

## Round 0.2 · accepted · Accept

Congratulations!
Thanks for your work to PeerJ.

Reviewer 2 ·

Basic reporting

Clear, unambiguous, professional English language used throughout.
Intro & background to show context.
 Literature well referenced & relevant.
 Structure conforms to PeerJ standards, discipline norm, or improved for clarity.
 Figures are relevant, high quality, well labelled & described.

Experimental design

Original primary research within Scope of the journal.
 Research question well defined, relevant & meaningful. It is stated how the research fills an identified knowledge gap.
 Rigorous investigation performed to a high technical & ethical standard.

Validity of the findings

All underlying data have been provided; they are robust, statistically sound, & controlled.
 Conclusions are well stated, linked to original research question & limited to supporting results.

Additional comments

no